# Mycotoxin Co-Occurrence in Milks and Exposure Estimation: A Pilot Study in São Paulo, Brazil

**DOI:** 10.3390/toxins13080507

**Published:** 2021-07-21

**Authors:** Matheus Frey, Roice Rosim, Carlos Oliveira

**Affiliations:** Department of Food Engineering, School of Animal Science and Food Engineering, University of São Paulo, Av. Duque de Caxias Norte, 225, Pirassununga CEP 13635-900, SP, Brazil; matheus.frey@usp.br (M.F.); roice@usp.br (R.R.)

**Keywords:** aflatoxins, ochratoxin A, fumoniins, zearalenone, trichothecenes, fluid milk, occurrence, exposure assessment

## Abstract

The aim of this study was to conduct a first evaluation on the co-occurrence of aflatoxins (AF) M_1_, B_1_, B_2_, G_1_ and G_2_; fumonisins (F) B_1_ and B_2_; deoxynivalenol (DON); de-epoxydeoxinivalenol (DOM-1); ochratoxin A (OTA); zearalenone (ZEN); α-zearalenol (α-ZEL); and β-zearalenol (β-ZEL) in 68 samples of fluid milk consumed in Pirassununga, São Paulo, Brazil. The probable daily intake (PDI) was also calculated for each mycotoxin evaluated. Mycotoxins were determined by liquid chromatography coupled to mass spectrometry. Sixty-two (91.2%) samples contained at least one type of mycotoxin. AFM_1_ was found in 6 samples (8.8%), and none of them presented concentrations above the Brazilian maximum permitted level in milk (500 ng/L). Low levels of non-regulated mycotoxins DOM-1, OTA, FB_1_, FB_2_, α-ZEL and β-ZEL were found in 6 (8.8%), 17 (25%), 10 (14.7%), 3 (4.4%), 39 (57.4%) and 28 (41.2%) samples of milk, respectively. None of the PDIs calculated for the quantified mycotoxins were above recommended values, indicating low exposure through milk consumption in the area studied. However, 21 samples (30.9%) contained 2–4 types of mycotoxins, which warrants concern about the potential adverse effects of mycotoxin mixtures in milks.

## 1. Introduction

Mycotoxins are toxic compounds produced as secondary metabolites by certain groups of fungi during their growth in food and feed products, which can lead to several toxic effects in animals and humans [1]. The most important fungal genera that produce mycotoxins are *Aspergillus*, *Penicillium* and *Fusarium*, and the main classes of mycotoxins produced by these genera are the aflatoxins (AF), ochratoxin A (OTA), fumonisins (FB), deoxynivalenol (DON) and zearalenone (ZEN) [2]. In adequate conditions of temperature and humidity, toxigenic fungi may contaminate foodstuffs in different phases of production and processing [3]. Moreover, when animals ingest these toxins, their metabolites or unmetabolized compounds may be transferred to products such as milk [4]. Therefore, dairy cows may shed mycotoxins and their respective biotransformation products in milk and milk products, leading to an additional source of dietary mycotoxins exposure for humans [5]. Several studies have addressed the interactive effects between mycotoxins such as AFs and OTA or FBs in animal models, including piglets, sheep and rats [6]. However, so far, the consequences of human exposure to multiple mycotoxins in the diet have not been elucidated [2].

Several mycotoxins may be excreted in milk either in their original forms or as biotransformed structures. However, AFM_1_ is the main compound that may be found in milk from lactating animals after ingesting feed contaminated with AFB_1_ [7]. Both AFB_1_ and AFM_1_ are hepatotoxic, teratogenic, immunosuppressive and classified as Group 1 carcinogens by the International Agency for Research on Cancer [8]. OTA is a nephrotoxic mycotoxin belonging to Group 2B, i.e., a probable carcinogen in humans [9]. In ruminants, OTA is rapidly degraded by microorganisms in the rumen to ochratoxin α (OTα) and phenylalanine [10]. Among the 28 structurally related FBs, FB_1_ is the most predominant form produced in nature [1]. FB_1_ exhibits hepatotoxic and nephrotoxic effects in several animal models, and is also in Group 2B [8]. DON has been placed in Group 3 by the International Agency for Research on Cancer [2]. However, this mycotoxin leads to vomiting and reduced weight gain due to food aversion, and problems in the regulation of the immune system [11]. De-epoxy-deoxynivalenol (DOM-1) is one of different metabolites that can be produced after the metabolization of DON in the upper gastrointestinal tract by anaerobic bacteria [12]. ZEN causes estrogenic abnormalities and generative syndromes in swine, being metabolized in the intestine or liver, where it generates α-zearalenol (α-ZEL) and β-zearalenol (β-ZEL) metabolites [13]. All these parent compounds or related metabolites can be excreted in urine. They are useful as exposure biomarkers for dietary mycotoxins in humans [14]. However, information on their occurrence in milk from lactating animals is scarce.

Considering the risks to human health posed by mycotoxins, exposure assessments to these dietary contaminants are critical, especially in fluid milk [15]. In addition, determining the level of human exposure may show the results of efforts in the area control of food contamination control—in particularly the efficacy of regulations that determine maximum permitted levels (MPL) for mycotoxins. In Brazil, the MPL values for AFM_1_ in milk and cheeses are 500 ng/mL and 2500 ng/kg, respectively [16]. Previous studies conducted in the past 10 years have indicated high frequencies with low levels of AFM_1_ in fluid milk produced in the country [15,17,18]. However, there are no regulations or available data on the occurrence of other mycotoxins potentially found in milk and milk products. Therefore, the objective of the present study was to conduct a preliminary assessment on the co-occurrence of major mycotoxins and some of their metabolites in pasteurized and ultra-high-temperature (UHT) milks consumed in the state of São Paulo, Brazil. Probable daily intakes (PDIs) for the mycotoxins analyzed were also determined based on the level of occurrence of said mycotoxins in milk samples, and the consumption data of fluid milk for Brazilian children and adults.

## 2. Results

Table 1 presents the occurrence levels of mycotoxins in grade A and regular pasteurized milks, and UHT milk consumed in Pirassununga, SP, Brazil. From the 68 samples analyzed, 62 (91.2%) presented quantifiable concentrations for one or more types of mycotoxins analyzed. AFB_1_, AFB_2_, AFG_1_, AFG_2_, DON and ZEN were not detected in any sample. AFM_1_ was quantified in 6 milk samples (8.8%) at levels ranging from 15–227 ng/mL. However, none of the samples analyzed had AFM_1_ concentrations higher than the Brazilian MPL adopted for this mycotoxin (500 ng/L) [16]. In addition, low levels of DOM-1, OTA, FB_1_, FB_2_, α-ZEL and β-ZEL were found in 6 (8.8%), 17 (25%), 10 (14.7%), 3 (4.4%), 39 (57.4%) and 28 (41.2%) samples of milk, respectively. There are no MPLs established for these mycotoxins in milk in Brazil.

The co-occurrence of mycotoxin residues in milk samples consumed in Pirassununga, state of São Paulo, Brazil is shown in Table 2. Twenty-one samples (30.9%) contained 2–4 types of mycotoxins. AFM_1_ or OTA in combination with ZEN metabolites (α- and/or β-ZEL) were the most frequent co-occurring mycotoxins in milk samples (*n* = 9).

The PDI values of mycotoxins in milks consumed by adults and children at different sampling times in the studied area are presented in Table 3. PDI was calculated using the standard body weights of 60 and 20 kg for adults and 5-year-old children, respectively, and the mean intake of 34.7 mL of milk per day [19]. It may be observed that PDI for AFM_1_ in 5-year-old children was considerably higher in the first sampling (0.128 ng/kg body weight/day), compared with the second one (0.029 ng/kg body weight/day).

## 3. Discussion

AFM_1_ was found in 6 samples of milk analyzed in the study (8.8%, Table 1), and none of them exceeded the MPL of 500 ng/L established in Brazil for AFM_1_ in fluid milk [16]. However, compared with the limit determined by European regulations (50 ng/L) [20], one milk sample showed an AFM_1_ concentrations above this MPL. The highest AFM_1_ level (227 ng/L) was found in one sample of UHT milk containing only this mycotoxin. This value is similar to the mean concentration described for UHT milk (196 ng/L) by Silva et al. [18] in Maringá, Paraná state, although the authors reported a much higher frequency of positive samples (87.5%). Jager et al. [21] collected milk samples in the city of Pirassununga and found 40% of them contaminated by AFM_1_. Contamination levels ranged from 9 to 69 ng/L, which is close to the values reported here. Previously, Shundo et al. [22] also described higher prevalence (70%) but low levels of AFM_1_ in UHT samples collected in the city of São Paulo. In the present study, the opposite was true: AFM_1_ was detected more frequently in pasteurized milk (12.5%), compared with UHT milk (3.1%). However, considering the incidence of all mycotoxins, UHT milk was more frequently contaminated (*n* = 65) than pasteurized milk (*n* = 59).

The most commonly co-occurring mycotoxins found in milk samples were AFM_1_ or OTA in combination with ZEN metabolites (α-ZEL and/or β-ZEL) (Table 2). There is no previous report on the occurrence of these mycotoxins in cow’s milk in Brazil. No available study on the excretion of DOM-1 in milk from dairy cows was found in the literature, either. However, a high frequency (91.7%) and a high mean level (268 ng/L) of OTA were reported in milks commercially available in Beijing, China [23]. Hence, our study confirms previous evidence indicating that OTA can be excreted in cow’s milk despite the detoxification by microorganisms in the rumen [10]. Furthermore, Gazzoti et al. [24] conducted a preliminary study on the occurrence levels of FB_1_ in milk and found a higher percentage of contaminated samples (80%) than the value obtained in this study (14.7%). However, the mean level reported (260 ng/L) by the authors was lower than described here (1.613 ng/mL). Although ZEN was not detected in milk samples in the present study, the occurrence levels of α-ZEL and β-ZEL also indicates probable high exposure of dairy cows to ZEN in feedstuffs. Our results for α-ZEL are in agreement to those described by Huang et al. [23], who observed mean levels of 36.7 ng/L of this ZEN metabolite in 41.7% of milk samples collected in Beijing, China. In our study, four and three samples contained three and four types of mycotoxins, respectively, which warrants concern about the potential adverse effects of mycotoxin mixtures in milks, especially in children, due to possible synergistic interactions of multiple toxic effects [2].

The frequencies and levels of AFM_1_ and *Fusarium* toxins found in milk in the present study are consistent with data reported previously indicating that AFs, FBs, ZEN and DON are the main mycotoxins detected in animal feed in Brazil [25,26]. However, there is no available information on the occurrence levels of OTA in feed for dairy cows in Brazil. Importantly, the co-occurrence of two or more mycotoxins was demonstrated in 51% of feed for poultry and dairy cows collected in small-scale farms from the states of São Paulo and Santa Catarina [26], which is in agreement with the data described in the present study. Thus, the co-occurrence of *Fusarium* toxins such as FBs, DON and ZEN and their related metabolites with AFM_1_ or OTA in milk as reported here stress the need for preventive measures to avoid the contamination of feed of dairy cows with these mycotoxins.

As for the PDI for AFM_1_ in this study (Table 3), the value for children was much higher in the first sampling (0.128 ng/kg body weight/day) than in the second one (0.029 ng/kg body weight/day). However, these results are lower than those reported by Shundo et al. [22] regarding the PDI in children from São Paulo city (0.23 ng/kg body weight /day). The PDI of AFM_1_ estimated for adults by Jager et al. [21] in Pirassununga (0.1 ng/kg body weight/day) was also higher than the value reported here (0.010–0.043 ng/kg body weight/day). Our data indicate that exposure to AFM_1_ through milk consumption have decreased in recent years in the area studied.

Regarding the other mycotoxins found in milk, there is no previous information on the potential contribution of cow’s milk for the human exposure. In a study carried out in European countries, Vidnes et al. [27] concluded that the PDI of ZEN in milk ranged from 12.0 to 29.3 ng/kg body weight/day in children, and from 2.1 to 4.8 ng/kg body weight/day in adults. These PDI values are much higher than those described for ZEN metabolites (α- and β-ZEL) in our study. Moreover, none of the PDI values observed in the present work (range: 0.267–2.947 ng/kg body weight/day) was greater than the tolerable daily intake (TDI) for ZEN and its metabolites (250 ng/kg body weight/day) recommended by the European Food Safety Authority [28]. Similarly, the PDI values obtained in our study for OTA (range: 0.030–0.184 ng/kg body weight/day), DOM-1 (range: 0.185–0.852 ng/kg body weight/day) and FBs (range: up to 2.921 ng/kg body weight/day) were lower than their respective TDI values: OTA = 14.3 ng/kg body weight/day [29], DON = 1000 ng/kg body weight/day [30] and FBs = 200 ng/kg body weight/day [31].

## 4. Conclusions

Results of this preliminary trial indicated low occurrence levels of AFM_1_ in milk samples consumed in São Paulo, Brazil. However, quantifiable levels of DOM-1, OTA, FB_1_, FB_2_, α-ZEL and β-ZEL were observed for the first time in Brazilian fluid milks. Sixty-two samples (91.2%) were contaminated with low concentrations of one or more types of mycotoxin, with 21 samples (30.9%) containing two-four types of mycotoxins. Despite high frequencies of positive samples, none of the PDIs calculated for adults and children exceeded the TDI values recommended for these mycotoxins. Further studies are necessary to evaluate the significance of co-occurring mycotoxins in milk considering a broad sampling approach in Brazil and the potential adverse effects of mycotoxin mixtures, especially in high milk consumers such as children.

## 5. Materials and Methods

### 5.1. Sampling Design

Samples were purchased in supermarkets in the city of Pirassununga, state of São Paulo, Brazil, at four different sampling times from October 2019 to June 2020. In each sampling time, samples of regular pasteurized milk (*n* = 1), grade A pasteurized milk (*n* = 8), and UHT milk (*n* = 8) were collected, totaling 68 samples at the end of the study. In each sampling, different milk brands and batches were collected. As regular pasteurized milk is not so commonly found in the market in Brazil, only a single sample per sampling was analyzed. Samples were collected in their 1-L original packaging and transported in isothermal boxes in dry ice directly to the laboratory. Upon arrival, all samples were divided into two 125-mL fractions and placed in 150-mL flasks to be stored at −18 °C until analysis.

### 5.2. Mycotoxin Analysis of Milk Samples

Samples (one fraction per sample) were analyzed using an in-house validated method [32] originally described by Flores-Flores and González-Peñas [33], with minor modifications. One mL of milk from a 125-mL fraction was transferred to a Falcon tube to which 4 mL of acetonitrile with 2% formic acid was added. Samples were stirred in a rotating mixer at 250 rpm for 15 min. and centrifuged at 2400× *g* for 10 min. After that, 4 mL of the supernatant were transferred to new Falcon tubes containing 60 mg of sodium acetate. Tubes were stirred again for 10 min. and centrifuged once more at 2400× *g* for 10 min. After the second centrifugation, 3.2 mL of acetonitrile (upper phase) were collected and filtered through a PVDF membrane (13 mm; 0.45 µm) in a 4-mL vial. The filtered portion was evaporated at 45 °C and, after drying, 200 µL of water:acetonitrile (95:5) solution containing 5 mM ammonium acetate and 0.1% acetic acid were added and immediately stirred in a vortex. Then, the contents of the vials were transferred to Eppendorf tubes, centrifuged again at 2400 × *g* for 10 min., and placed in glass inserts.

Analyses were carried out using an ultra-performance liquid chromatography (Acquity I-Class^®^, Waters, Milford, MA, USA) coupled to a tandem mass spectrometer (Xevo TQS^®^, Waters, Milford, MA, USA) and electrospray ionization. Individual mycotoxin standard solutions and calibration curves were prepared strictly following the procedures recommended by Coppa et al. [32]. A work solution with mixed mycotoxins was prepared in water: acetonitrile (50:50), containing AFM_1_, AFB_1_, AFB_2_, AFG_1_, AFG_2_, OTA, FB_1_, FB_2_, ZEN, α-ZEL and β-ZEL at 100 ng/mL, and DON and DOM-1 at 750 ng/mL. This solution was used to prepare five matrix-matched calibration standards at the range levels expressed in Table 4. Five 5 μL of the extracts and standards were injected using gradient elution in a mobile phase made up of water (eluent A) and acetonitrile (eluent B), both containing 5 mM ammonium acetate and 0.1% acetic acid and kept at 0.6 mL/min, as described elsewhere [32]. Total chromatography run for each sample was 10 min. The mass spectrometer was operated in MRM mode (Multiple Reaction Monitoring), with main parameters as described in Table 4. Limits of detection (LOD) and quantification (LOQ) were determined considering signal: noise ratios of 1:3 and 1:10, respectively.

### 5.3. Estimation of Probable Daily Intake of Mycotoxins

From the results obtained in the analyses, PDI was calculated based on Equation (1) [21].
PDI = (C_M_ Mycotoxin × DI_M_)/BW_M_(1)

C_M_ = Mean concentration of the mycotoxin in milk samples (μg/kg);

DI_M_ = Mean daily intake of fluid milk estimated in Brazil (kg) [19];

BW_M_ = Mean individual body weight estimated for Brazilian adults and 5-year-old children (kg) [34].

### 5.4. Statistical Analysis

PDI values were analyzed using a non-parametric Mann-Whitney test, considering positives samples for mycotoxins in milk samples (containing levels above LOQ) and *p* < 0.05 [35].

## Figures and Tables

**Table 1 toxins-13-00507-t001:** Occurrence of mycotoxins in grade A milk, pasteurized milk, and ultra-high-temperature (UHT) milk consumed in Pirassununga, SP, Brazil.

	Type A Milk (*n* = 32)		Pasteurized Milk (*n* = 4)		UHT Milk (*n* = 32)		Total (*n* = 68)	
	*n* (%)	Mean (Range) (ng/L)	*n* (%)	Mean (Range) (ng/L)	*n* (%)	Mean (Range) (ng/L)	*n* (%)	Mean (Range) (ng/L)
AFM_1_	4 (12.5)	19 (15–21)	1 (25)	32	1 (3.1)	227	6 (8.8)	93 (15–227)
DOM-1	3 (9.4)	386 (313–459)	0	<LOQ	3 (9.4)	448 (327–682)	6 (8.8)	417 (313–682)
OTA	6 (18.7)	78 (43–201)	1 (25)	73	10 (31.2)	68 (43–148)	17 (25)	72 (43–201)
FB_1_	4 (12.5)	1609 (1465–1734)	2 (50)	1775 (1595–1955)	4 (12.5)	1536 (1393–1705)	10 (14.7)	1613 (1393–1955)
FB_2_	2 (6.3)	481 (464–499)	1 (25)	1297	0	<LOQ	3 (4.4)	753 (464–1297)
α-ZEL	15 (46.9)	608 (416–966)	1 (25)	679	23 (71.9)	612 (334–936)	39 (57.4)	612 (334–966)
β-ZEL	16 (50)	1514 (445–3146)	1 (25)	2326	11 (34.4)	0.982 (332–1949)	28 (41.2)	1334 (332–3146)

*n*: Number of samples containing levels higher than the limit of quantification (LOQ). See Section 5.2 for the LOQ of each mycotoxin. LOQ: Limit of quantification; AFM1: aflatoxin M1; DOM-1: de-epoxy-deoxynivalenol; OTA: ochratoxin A; FB: fumonisin; α-ZEL: α-zearalenol; β-ZEL: β-zearalenol.

**Table 2 toxins-13-00507-t002:** Mycotoxin co-occurrence in milk samples consumed in Pirassununga, state of São Paulo, Brazil.

Types of Co-Occurring Mycotoxins	Number of Samples Positive for Mycotoxins (*n* = 68) ^a^
AFM_1_, ZEL	4
OTA, ZEL	5
FB, OTA	2
DOM-1, ZEL	2
FB, ZEL	1
AFM_1_, DOM-1, ZEL	1
AFM_1_, FB, ZEL	1
FB, DOM-1, ZEL	1
FB, OTA, ZEL	1
AFM_1_, FB, DOM-1, ZEL	1
FB, OTA, DOM-1, ZEL	2
Total (%)	21 (30.9)

^a^ Samples containing levels higher than the limit of quantification (LOQ), see Section 5.2 for LOQ of each mycotoxin. AFM_1_: aflatoxin M_1_; DOM-1: de-epoxy-deoxynivalenol; OTA: ochratoxin A; FB: fumonisin (B_1_ and/or B_2_); ZEL: zearalenol (α- and/or β- isomers).

**Table 3 toxins-13-00507-t003:** Mean probable daily intake (PDI) of mycotoxins in milks consumed by adults and 5-year-old children at different sampling times in Pirassununga, state of São Paulo, Brazil.

Mycotoxins	Mean PDI (ng/kg Body Weight /Day) ^1^							
	October/2019		January/2020		May/2020		June/2020	
	Adults	Children	Adults	Children	Adults	Children	Adults	Children
AFM_1_	0.043 ^b^	0.128 ^a^	0.010 ^b^	0.029 ^b^	0	0	0	0
DOM-1	0.284 ^b^	0.852 ^a^	0.224 ^b^	0.673 ^a^	0.185 ^b^	0.555 ^a^	0	0
OTA	0.058 ^c^	0.173 ^a^	0.061 ^c^	0.184 ^a^	0.030 ^c^	0.090 ^b^	0.035 ^c^	0.104 ^b^
FB_1_	0.974 ^b^	2.921 ^a^	0.953 ^b^	2.860 ^a^	0.876 ^b^	2.629 ^a^	0.915 ^b^	2.746 ^a^
FB_2_	0.750 ^b^	2.250 ^a^	0.268 ^c^	0.805 ^b^	0	0	0.289 ^c^	0.866 ^b^
α-ZEL	0.468 ^c^	1.403 ^a^	0.340 ^c^	1.020 ^b^	0.317 ^c^	0.951 ^b^	0.319 ^c^	0.957 ^b^
β-ZEL	0.982 ^c^	2.947 ^a^	0.957 ^c^	2.871 ^a^	0.509 ^d^	1.526 ^b^	0.267 ^d^	0.800 ^c^

^1^ Calculated according to the equation: PDI (ng/kg bw/day) = [Mean concentration (ng/mL) × daily intake (mL)]/body weight (kg). ^a–d^ In the same row, means followed by different superscripts are significantly different (*p* < 0.05). AFM_1_: aflatoxin M_1_; DOM-1: de-epoxy-deoxynivalenol; OTA: ochratoxin A; FB: fumonisin; α-ZEL: α-zearalenol; β-ZEL: β-zearalenol.

**Table 4 toxins-13-00507-t004:** Analytical parameters of the method for determination of mycotoxins in milk.

Mycotoxin	RT (min.)	Mass (g/mol)	Molecularion	Transition (*m/z*)	Calibration Range (ng/L)	LOD (ng/L)	LOQ (ng/L)
AFM_1_	4.09	328.3	[M+H]^+^	329.0 > 273.1 ^a^	31–500	5	15
				329.0 > 229.0 ^b^			
AFB_1_	4.80	312.3	[M+H]^+^	312.7 > 284.9 ^a^	31–500	40	120
				312.7 > 241.1 ^b^			
AFB_2_	4.59	314.3	[M+H]^+^	314.7 > 259.0 ^a^	31–500	40	120
				314.7 > 287.0 ^b^			
AFG_1_	4.46	328.3	[M+H]^+^	328.9 > 243.0 ^a^	63–1000	40	120
				328.9 > 199.5 ^b^			
AFG_2_	4.18	330.3	[M+H]^+^	330.9 > 245.0 ^a^	94–1500	60	180
				330.9 > 188.9 ^b^			
DON	2.04	296.3	[M+H]^+^	297.3 > 249.1 ^a^	313–5000	333	1223
				297.3 > 231.1 ^b^			
DOM-1	2.62	282.1	[M+Ac]^−^	339.2 > 249.1 ^a^	625–10,000	170	310
				339.2 > 59.1 ^b^			
OTA	5.70	403.1	[M+H]^+^	404.0 > 238.9 ^a^	63–1000	5	17
				404.0 > 357.9 ^b^			
FB_1_	5.34	721.8	[M+H]^+^	722.5 > 334.0 ^a^	625–10,000	300	620
				722.5 > 352.1 ^b^			
FB_2_	5.74	705.8	[M+H]^+^	706.5 > 336.2 ^a^	625–10,000	100	330
				706.5 > 318.3 ^b^			
ZEN	6.01	318.1	[M−H]^−^	317.1 > 175.1 ^a^	313–5000	100	300
				317.1 > 130.9 ^b^			
α-ZEL	5.53	320.2	[M–H]^−^	319.1 > 275.2 ^a^	313–5000	73	283
				319.1 > 160.2 ^b^			
β-ZEL	5.76	320.2	[M–H]^−^	319.1 > 275.2 ^a^	313–5000	60	200
				319.1 > 160.2 ^b^			

RT: Retention time; LOD: Limit of detection; LOQ: Limit of quantification; AF: aflatoxin; DON: deoxynivalenol; DOM-1: de-epoxy-deoxynivalenol; OTA: ochratoxin A; FB: fumonisin; ZEN: zearalenone; α-ZEL: α-zearalenol; β-ZEL: β-zearalenol. ^a^ Transitions used in the quantification. ^b^ Transitions used in the confirmation.

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
