# Peer review of "Mycotoxin Co-Occurrence in Milks and Exposure Estimation: A Pilot Study in São Paulo, Brazil"

_toxins, 2021, doi:10.3390/toxins13080507_

Round 1
Reviewer 1 Report
The article “Mycotoxin co-occurrence in milks and exposure estimation: a pilot study in São Paulo, Brazil” aims at identify and quantify the mycotoxins presence in pasteurized and UHT milks sampling in the city Pirassununga in Brazil at four different sampling times. The authors found UHT milk was more frequently contaminated than pasteurized milk and the most frequent co-occurring mycotoxins in milk samples were AFM1 or OTA in combination with ZEN metabolites.
In general, I find the manuscript is comprehensively described and it is well organized. This work fits the scope of the journal and Special Issue "Mycotoxin Contamination and Food Safety". But, in my opinion, this manuscript is not classifiable as “Research Article”, but as “Communication”. In fact, no experimental data are shown but results of monitoring of occurrence mycotoxin in milk samples.
I think this type of work is important because monitoring activities can contribute effectively to the control of the mycotoxins problem to reduce risk of disease in population.
Other minor comments:
Please, In the text, use reference numbers placed in square brackets [ ]; for example [1], [1–3] or [1,3]. See “Instruction for Authors” (https://www.mdpi.com/journal/toxins/instructions).
Line 136: please, write 1 as subscript number “FB1”
Author Response
The authors thank the constructive comments and suggestions of the Reviewers for improving the manuscript. We have addressed all the concerns regarding the manuscript and included below a point-by-point response to the Reviewers. In addition, the changes done in the text were highlighted in yellow in the new, revised version of the manuscript.
Reviewer #1: The article “Mycotoxin co-occurrence in milks and exposure estimation: a pilot study in São Paulo, Brazil” aims at identify and quantify the mycotoxins presence in pasteurized and UHT milks sampling in the city Pirassununga in Brazil at four different sampling times. The authors found UHT milk was more frequently contaminated than pasteurized milk and the most frequent co-occurring mycotoxins in milk samples were AFM1 or OTA in combination with ZEN metabolites.
Answer: Thanks for summarizing the main findings of the study.
In general, I find the manuscript is comprehensively described and it is well organized. This work fits the scope of the journal and Special Issue "Mycotoxin Contamination and Food Safety". But, in my opinion, this manuscript is not classifiable as “Research Article”, but as “Communication”. In fact, no experimental data are shown but results of monitoring of occurrence mycotoxin in milk samples.
I think this type of work is important because monitoring activities can contribute effectively to the control of the mycotoxins problem to reduce risk of disease in population.
Answer: Thanks for your valuable and positive comments. The type of article was changed as “Communication”, as suggested.
Other minor comments:
Please, In the text, use reference numbers placed in square brackets [ ]; for example [1], [1–3] or [1,3]. See “Instruction for Authors” (https://www.mdpi.com/journal/toxins/instructions).
Answer: The citations were replaced with their respective numbers in square brackets, according to the journal’s guidelines.
Reviewer 2 Report
eg- line 46 in manuscript authors put two times and Also in my opinion, Introduction section, where authors discuss about side effects of mycotoxins could be detailed, including aspects concerning interactions between mycotoxins in human/animal bodyAuthor Response
The authors thank the constructive comments and suggestions of the Reviewers for improving the manuscript. We have addressed all the concerns regarding the manuscript and included below a point-by-point response to the Reviewers. In addition, the changes done in the text were highlighted in yellow in the new, revised version of the manuscript.
Reviewer #2: eg- line 46 in manuscript authors put two times and Also in my opinion, Introduction section, where authors discuss about side effects of mycotoxins could be detailed, including aspects concerning interactions between mycotoxins in human/animal body.
Answer: Thanks for the comments. The citation in L.46 was replaced with its number, according to the journal’s guidelines. The Introduction section was amended with details on the toxic effects of mycotoxins and information on the potential interactions between mycotoxins, as suggested.
Reviewer 3 Report
Please follow the journal guidelines for Abstract and References. Specific comments were directly made in the manuscript which is attached. Portions of sentences have been either highlighted or struck off and edited.

Author Response
The authors thank the constructive comments and suggestions of the Reviewers for improving the manuscript. We have addressed all the concerns regarding the manuscript and included below a point-by-point response to the Reviewers. In addition, the changes done in the text were highlighted in yellow in the new, revised version of the manuscript.
Reviewer #3: Please follow the journal guidelines for Abstract and References. Specific comments were directly made in the manuscript which is attached. Portions of sentences have been either highlighted or struck off and edited.
Answer: Thanks a lot for the comments and suggestions made in the PDF of the submission. The Abstract and References were adjusted according to the journal’s guidelines, as suggested.